# Mental Health and the SARS-COV-2 Epidemic—Polish Research Study

**DOI:** 10.3390/ijerph17197015

**Published:** 2020-09-25

**Authors:** Monika Talarowska, Jan Chodkiewicz, Natalia Nawrocka, Joanna Miniszewska, Przemysław Biliński

**Affiliations:** 1Institute of Psychology, University of Lodz, 91-433 Lodz, Poland; jan.chodkiewicz@uni.lodz.pl (J.C.); natalia.nawrocka2@unilodz.eu (N.N.); joanna.miniszewska@uni.lodz.pl (J.M.); 2The President Stanisław Wojciechowski State University of Applied Sciences in Kalisz, 92-800 Kalisz, Poland; bildom@gmail.com; 3Copernicus Memorial Multidisciplinary Comprehensive Cancer and Traumatology Center, 93-513 Lodz, Poland

**Keywords:** mental health, stress, COVID-19, GHQ-28

## Abstract

*Background*: The aim of this study was to assess the mental state of Poles in the first weeks of the SARS-COV-2 epidemic. *Methods*: In the study, the General Health Questionnaire-28 (GHQ-18), The Perceived Stress Scale (PSS-10), and Mini-Cope were used. *Results*: The study was conducted on a group of 443 individuals, including 348 women (78.6%) and 95 men (21.4%). There were more women (*χ*^2^ = 6.42, *p* = 0.02) in the group of people with high results in the GHQ-28 questionnaire and the differentiating factors between those with sten scores above 7 (significantly deteriorated mental health) and those with average or low results (sten score below 7) turned out to be: treatment for mental disorders before the pandemic (*χ*^2^ = 19.57, *p* < 0.001) and the use of psychotherapy during the pandemic (*χ*^2^ = 4.21, *p* = 0.04) and psychiatric pharmacotherapy (*χ*^2^ = 8.31, *p* = 0.01). The presence of suicidal thoughts since the appearance of the pandemic-related restraints and limitations significantly differentiates the compared groups (*χ*^2^ = 38.48, *p* < 0.001). *Conclusions*: Every fourth person in the examined group (over 26% of the respondents) recorded results that indicate a high probability of mental functioning disorders. Approximately 10% of the respondents signalled the occurrence of suicidal thoughts since the beginning of the pandemic. The respondents complain mainly about problems in everyday life, lack of satisfaction from one’s own activities, tension, trouble sleeping, and feelings of exhaustion. Individuals with significantly reduced mental well-being use non-adaptive coping strategies, such as denying problems, emotional discharge, taking substances, discontinuation of action, and blaming themselves for the situation. The risk factors for the deterioration of the mental state of the respondents during the pandemic include psychiatric treatment before the beginning of the pandemic, the presence of suicidal thoughts during forced isolation, and the use of non-adaptive coping strategies (denial of the existence of problems, emotional discharge, use of psychoactive substances, discontinuation of action, and blaming oneself for the situation).

## 1. Introduction

During the first three months of 2020, the world as we know it changed gradually from day to day, from week to week. The COVID-19 (Coronavirus Disease 2019) epidemic quickly evolved from a “local” disease into a fast-growing and fear-inducing global pandemic [1]. The situation resulting from the pandemic has all the characteristics of a collective trauma [2]. It appeared unexpectedly and touched all areas of life—family, work, and relationships with loved ones. Gradually, it reduced our usual activity, not allowing us to take care of ourselves in the usual way [3].

In such situations, a whole array of unpleasant emotions appear, including sadness, anxiety, anger, guilt or grief, and the feeling of losing control over one’s life. These emotions are natural stages of mourning that appear in the face of a loss (such a loss is undoubtedly the limitations that have lasted for several months) and were described many years ago by Elizabeth Kübler-Ross [4]. On the other hand, these emotions can be a symptom of mental disorders, such as episodes of depression or anxiety disorders [5].

A particularly difficult emotion associated with the epidemic situation is the feeling of uncertainty. It is unfavourable for the body since the stressor affecting us now is a completely new phenomenon. The lack of early warnings of imminent danger made it impossible to prepare for the upcoming situation and hindered initial healthy adaptation. At the moment, we know neither a cure to fight the virus effectively, nor the approximate time of when a vaccine will be available. The long-term health and social implications of the virus are unknown. We also do not know when (or if at all) life will return to its pre-pandemic state [6]. Ubiquitous uncertainty makes it difficult to plan for the future and thus, becomes a source of additional stress [6]. 

Figure 1 shows the factors that are a source of stress during the SARS-CoV-2 (severe acute respiratory syndrome coronavirus 2) epidemic and their possible psychological and social consequences [7].

The aim of this study was to assess the mental state of Poles in the first weeks of the SARS-COV-2 epidemic.

The presented article is a part of a longitudinal study conducted by a team from the Institute of Psychology at the University of Lodz, in which we evaluated, apart from mental state, the frequency of alcohol consumption in the studied group [8]. The presented results refer to the study carried out in April 2020. The second part of the study, scheduled for June and September 2020, will assess the long-term effects of the pandemic that we are still facing.

## 2. Methods

Due to the current epidemiological situation, the research—the results of which are presented herein—was conducted fully anonymously and online using a Google form. We used social media, primarily Facebook, to distribute the survey. The respondents (adults only) were obtained with the application of the “snowball” method (the participants sent each other a link to the survey). Participation in the study was free of charge for the respondents. Each of the participants had the opportunity to see the results of the tests performed.

The research procedure was conducted in accordance with the Declaration of Helsinki of the World Medical Association [9] and the ethical codes of the Belmont Report [10]. The study was accepted by the State Agency for the Prevention of Alcohol-Related Problems (PARPA) in Poland.

Selected methods of descriptive statistics and methods of statistical reasoning were used in the statistical analysis of the collected material. For non-parametric variables, the non-parametric distribution compliance test (χ^2^ test) was used for statistical comparisons between the studied groups; for parametric variables, the Student’s *t*-test was used [11]. All statistical calculations were conducted using STATISTICA PL computer software (version 13.3).

A self-reported questionnaire was used in the study to collect sociodemographic data and information related to the emotional state of the respondents. Other research methods included:**GHQ–28** (General Health Questionnaire-28 by D. Goldberg)—this method measures four aspects of mental health: somatic symptoms (e.g., headaches, exhaustion, weakness, subjective malaise), level of anxiety and insomnia, social impairment, and symptoms of depression [12,13]. The reliability of the method (Cronbach Alpha) in the presented study ranged from 0.79 (somatic symptoms) to 0.91 (depression). The reliability of the whole scale totalled 0.94.**PSS-10** (The Perceived Stress Scale)—this method allows assessing the intensity of stress related to one’s own life during the last month [14,15]. In the tested sample, the Cronbach Alpha for the method totalled 0.84.**MINI-COPE**—an estimation of the available ways of dealing with stress in the form of disposable and situational coping. In the presented analyses, the second option was used (coping with a pandemic situation). The internal conformity of the Polish version of the questionnaire is 0.86 [16,17].

## 3. Material

The study was conducted on a group of 443 individuals, including 348 women (78.6%) and 95 men (21.4%). The average age of the respondents was 31.9 years (*SD* = 1.31). The youngest person was 18, while the oldest participant was 68 years old. 

## 4. Results

The characteristics of the sociodemographic variables are presented in Table 1. Table 2 provides a description of the personal situation of the respondents during the pandemic.

Table 3 presents the collected data on the somatic and mental health of the examined individuals.

The results obtained in the GHQ-28 questionnaire and The Perceived Stress Scale in the studied group are presented in Table 4.

The results obtained in the two methods analysing the state of mental health of the respondents in recent weeks (GHQ-28 and PSS-10) were compared with the Polish standards created during their adaptation [13,15]. The total result in the study group for the GHQ-28 questionnaire is *M* = 32.74 (7 sten, high score). It means that the respondents assess their mental health as unsatisfactory.

A detailed analysis of the results obtained (Table 4) indicates the predominance of disorders in the daily functioning of the subjects (coping with duties, satisfaction with performing tasks), the presence of somatic symptoms (headaches, exhaustion, weakness, subjective malaise), and experiencing increased levels of anxiety and sleep problems.

When analysing the distribution of the obtained results, it was found that more than half of the respondents (52.82%) recorded results indicating a sten score of 7 or higher (high scores). It is exceedingly worrying that as many as 116 individuals (26.18%) recorded results corresponding to a sten score of 9 or 10, which argues in favour of the possibility of noticeable mental health problems in this group.

On the other hand, the average results obtained in The Perceived Stress Scale (PSS-10) in the study group are within the range of average results (6 sten), but a detailed analysis of the data obtained indicates (similarly to GHQ-28) that slightly more than half of the respondents (less than 53%) obtained results corresponding to a 7 sten score and higher, which also indicates high or very high levels of perceived stress.

In order to look at the correlation between mental health and the level of perceived stress and the strategies of pandemic stress management used by the respondents, the level of these variables was compared (Student’s *t*-test) in two groups, i.e., the group with high scores (sten score above 7) in the GHQ-28 questionnaire and the group with average or low scores in this test (sten score below 7 in GHQ-28) (Table 5).

As Table 5 shows, there are a number of statistically significant differences between the people characterised by a deteriorated state of mental health and those without any such symptoms. People with deteriorated mental health report significantly more severe stress in the last month (*Cohen’s d* = 1.70, strong effect). They are also less likely to cope with the current difficult situation by looking for positive aspects of it (positive reframing) and have greater difficulties in accepting it (acceptance). However, they more often look for instrumental support and use unsuitable ways of dealing with the situation, such as denying the situation, emotional discharge, taking psychoactive substances, discontinuation of actions, and blaming themselves for the situation (moderate effects—*Cohen’s d* of 0.38 to 0.59 and in the case of behavioral disengagement and self-blame, 0.76 and 0.91 (strong), respectively).

At the next stage of the analysis, we checked whether the sociodemographic variables and the selected clinical variables differentiate people with high (sten score above 7) and average or low (sten score below 7) results obtained in the GHQ-28 questionnaire.

There were more women *(χ*^2^ = 6.42, *p* = 0.02) in the group of people with high results in the GHQ-28 questionnaire and the differentiating factors between those with sten scores above 7 (significantly deteriorated mental health) and those with average or low results (sten score below 7) turned out to be: treatment for mental disorders before the pandemic (*χ*^2^ = 19.57, *p* < 0.001) and the use of psychotherapy during the pandemic (*χ*^2^ = 4.21, *p* = 0.04) and psychiatric pharmacotherapy (*χ*^2^ = 8.31, *p* = 0.01). Suicide attempts in the period before March 2020 do not differentiate the respondents (*χ*^2^ = 1.92, *p* = 0.165), while the presence of suicidal thoughts since the appearance of the pandemic-related restraints and limitations significantly differentiates the compared groups (*χ*^2^ = 38.48, *p* < 0.001).

The factors that did not differentiate between people with high (sten score above 7) and average or low (sten score below 7) results obtained in the GHQ-28 questionnaire were as follows: marital status (*χ*^2^ = 0.086, *p* = 0.822), having children (*χ*^2^ = 0.06, *p* = 0.792), level of education (*χ*^2^ = 4.52, *p* = 0.235), place of residence (*χ*^2^ = 4.85, *p* = 0.08), nature of employment (*χ*^2^ = 5.65, *p* = 0.226), current living conditions (*χ*^2^ = 1.37, *p* = 0.241), or possible quarantine (*χ*^2^ = 0.93, *p* = 0.334). Moreover, clinical variables, such as the presence of somatic disease, did not differentiate between the examined groups (*χ*^2^ = 0.03, *p* = 0.873).

## 5. Discussion

The American Psychiatric Association (APA) estimates that the negative impact of SARS-CoV-2 on the human psyche will be observed in nearly 50% of the population [18]. These symptoms may appear even several months after the lifting of the strictest limitations and restrictions [19]. A study conducted in the United States after the end of the 2009 Influenza A (H1N1) epidemic indicated a four times higher risk of post-traumatic stress symptoms in quarantined children compared to their non-quarantined peers [20]. In the case of the parents of the examined children, those indicators totalled, respectively, 28% to 6% [20]. Furthermore, after lifting the restrictions related to the SARS (severe acute respiratory syndrome) epidemic in 2003, the respondents reported symptoms from the mental sphere, such as avoidance of crowded places [26%] and public spaces [21%], combined with a feeling of anxiety for several weeks after the end of the epidemic (a study conducted in Canada) [21].

The results obtained in the study confirm data from earlier years. The mean score in the GHQ-28 questionnaire obtained in the studied group indicates a deteriorated mental state of the subjects as compared to the Polish standards. The respondents complain mainly about problems in everyday life, lack of satisfaction from one’s own activities, tension, trouble sleeping, and feelings of exhaustion. Individuals with significantly poorer mental wellbeing (sten scores higher than 7 in GHQ 28) experience stress more intensely than others and use rather unsuitable coping strategies (denying problems, emotional discharge, taking substances, giving up, and blaming themselves for the situation). It can be cautiously assumed that these individuals, who are most likely to experience high levels of anxiety, prefer passive counter-measures (“freezing”) as a behavioural (and endocrine) pattern in situations where danger is perceived as inevitable but relatively distant [22].

It is highly alarming that every fourth person (over 26%) in the surveyed group recorded results that indicate a high probability of mental health disorders and that about 10% of the surveyed individuals have signalled the occurrence of suicidal thoughts since the beginning of the pandemic. These results are not surprising. During the SARS epidemic in 2003, a 30% increase in suicide attempts was observed among people over 65 years of age and nearly 50% of people after the infection were found to have increased anxiety symptoms [23]. Similar relationships were found in the United States during the 1918–1919 influenza pandemic [24]. What is more, the suicidal risk may increase due to the increasing stigmatisation of patients infected with the SARS-CoV-2 virus, their family members [25], and representatives of medical services [21].

More than 30% of the respondents declared psychiatric treatment before the beginning of the pandemic; 9.3% of the respondents used psychotherapeutic assistance during forced isolation and 11.3% of the respondents used pharmacotherapy prescribed by a psychiatrist at that time. Moreover, these people declared significantly worse mental health compared to those who did not receive psychiatric or psychological assistance before March 2020 (sten scores above 7 in the GHQ-28 questionnaire). Therefore, the results should also be considered in the context of people treated due to mental disorders before the start of the pandemic [26]. This group is particularly exposed to the long-term emotional and somatic consequences of the current situation for two reasons, i.e., possible exacerbation of pre-existing mental symptoms as a consequence of the prolonged crisis and because of the postponed consequences of the applied pharmacological treatment (e.g., risk of metabolic syndrome symptoms) [27]. Many psychiatric patients are also addicted to nicotine, which is associated with all the physical burdens of this addiction and can worsen the prognosis for a SARS-CoV-2 infection [28]. Recent reports also indicate that COVID-19-associated acute respiratory distress syndrome can affect not only the brain, but can also trigger an increased immune response, which will have repercussions for the central nervous system, increasing the risk of mental symptoms and disorders [7].

A different group, yet also significantly related to the risk of emotional consequences of the pandemic, are representatives of medical services [29,30,31]. Yip et al. [23] estimate that emotional difficulties as a consequence of the SARS-CoV-2 pandemic will occur in almost 30% of health care workers. Similarly, during the SARS epidemic in Singapore, as many as 27% of medical service employees reported symptoms related to the mental sphere [32]. It should be stressed that this phenomenon concerns not only those people who are at the front line of the fight against the virus, but also those among us who are affected by the epidemic in terms of professional and organisational changes [33,34]. Future research should answer the question about the mental functioning of this group. 

When analysing the results obtained, it is also worthwhile to refer to research on the relationship between crisis situations and the mental health of individuals. The results of research on the correlation between crisis and mental health based on the 2008 global economic crisis indicate that the crisis is an important and highly significant factor that has a negative impact on mental health [35]. A rise in unemployment, increased workload, staff reduction, and wage reductions are associated with an increased frequency of mood disorders, anxiety, depression, dystonia, and suicide [35]. It seems obvious that the impact of the current crisis may be similar, which will require further studies and the search for public health counterfactors on their basis.

In an attempt to determine the risk factors for the deterioration of the emotional state of the respondents during the SARS-CoV-2 pandemic, one can mention: psychiatric treatment before the beginning of the pandemic, the presence of suicidal thoughts during forced isolation, and the use of non-adaptive coping strategies (denial of the existence of problems, emotional discharge, use of psychoactive substances, discontinuation of action, and blaming oneself for the situation). This seems to be a highly significant action, applying in particular to the younger demographics within the society, such as in Poland. Research carried out in this country indicates so [36,37]. Such actions should be global in nature, similar to cases of other illnesses [38,39].

It should be stressed that the presented study has some limitations. It is a cross-sectional study based on a Polish population. Mainly women living in large cities, with secondary or higher education, reported for the study. Moreover, due to its nature (the snowball sampling), the study was conducted only on the individuals who use information and communication technologies.

However, it should be remembered that the presented study is one of the few conducted in Poland during the SARS-CoV-2 pandemic. The study relates directly to the assessment of the mental state of the respondents. A large group of people participated in the study and the diagnostic methods that were used in the study are widely known.

Human beings feel comfortable and safe when, on the basis of incoming information, they are able to recognise, predict, and control, to a large extent, the world around them and subsequent events [40]. The coronavirus pandemic deprives the world of the comfort of relative predictability and the sense of control over life in many of its spheres. For most people, this is the first such experience which, like any change, requires gradual adaptation [40].

Referring to the concept of Hans Selye’s general adaptation syndrome, we can say that most of us are currently in the adaptation phase [41]. This condition is the body’s natural physical and emotional response to the stimuli. According to H. Selye’s theory, in case of a prolonged stressful situation—even if its strength is not too great—our body’s resources will drastically decrease [41], which will eventually result in the occurrence of somatic and/or mental symptoms [22,42]. The second part of the research can provide interesting findings in this area.

## 6. Conclusions

Every fourth person in the examined group (over 26% of the respondents) recorded results that indicate a high probability of mental functioning disorders.Approximately 10% of the respondents signalled the occurrence of suicidal thoughts since the beginning of the pandemic.The respondents mainly complained about problems in everyday life, lack of satisfaction from one’s own activities, tension, trouble sleeping, and feelings of exhaustion.Individuals with significantly reduced mental well-being use non-adaptive coping strategies, such as denying problems, emotional discharge, taking substances, discontinuation of action, and blaming themselves for the situation.The risk factors for the deterioration of the mental state of the respondents during the pandemic include psychiatric treatment before the beginning of the pandemic, the presence of suicidal thoughts during forced isolation, and the use of non-adaptive coping strategies (denial of the existence of problems, emotional discharge, use of psychoactive substances, discontinuation of action, and blaming oneself for the situation).

## Figures and Tables

**Figure 1 ijerph-17-07015-f001:**
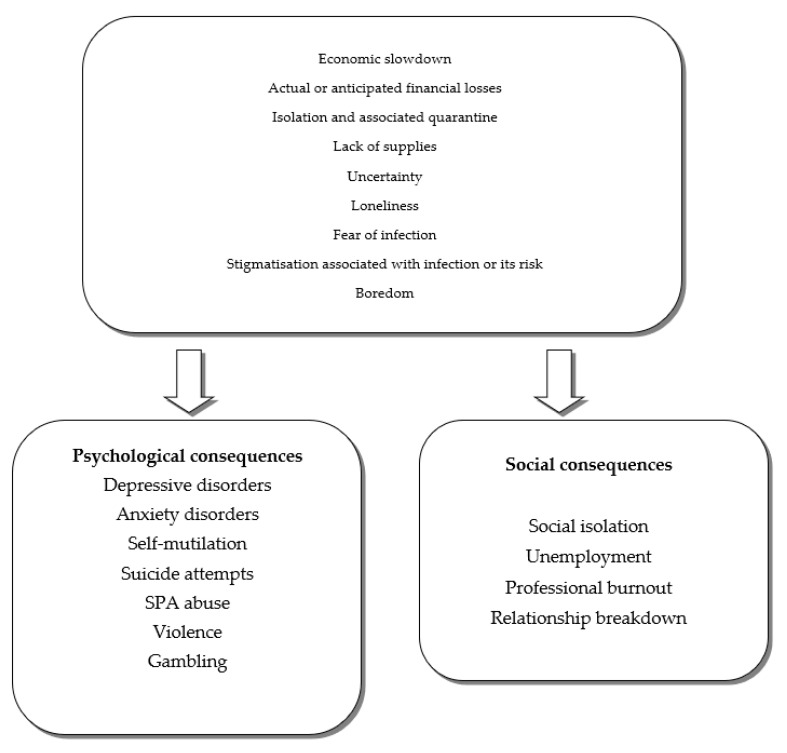
The factors that are a source of stress during the SARS-CoV-2 epidemic and their possible psychological and social consequences [7].

**Table 1 ijerph-17-07015-t001:** Characteristics of the examined group—sociodemographic variables (*N* = 443).

Sociodemographic Variables	*N* = 443	%
Place of residence		
Rural settlement	69	15.58
City with up to 100,000 inhabitants	100	22.57
City with more than 100,000 inhabitants	274	61.85
Education		
Primary/lower secondary education	8	1.81
Vocational	4	0.90
Secondary	170	38.37
Higher	261	58.92
Marital status		
Single	210	47.40
Married, civil partnership	204	46.05
Divorced, legal separation	27	6.09
Widow/widower	2	0.45
Having children		
Yes	140	31.60
No	303	68.40
Employment		
I study/learn	93	20.99
I study and work	79	17.83
I am a full-time employee	225	50.79
I work occasionally	21	4.74
I do not work	25	5.64

**Table 2 ijerph-17-07015-t002:** Daily functioning of the respondents during the pandemic (*N* = 443).

Daily Functioning of the Respondents	*N* = 443	%
Current residence/work conditions:		
I stay at home and I do not go outside at all	53	12
I stay at home and go out occasionally (walk, shopping)	290	65.5
I stay at home, but I go to work regularly	88	19.9
I go outside (friends, family)	12	2.7
When filling in the test, are you or have you been in quarantine for two weeks?		
Yes	38	8.6
No	405	91.4

**Table 3 ijerph-17-07015-t003:** Somatic and mental health of the examined individuals (*N* = 443).

Somatic and Mental Health of the Examined Individuals	*N* = 443	%
Are you or have you been suffering within the last year from any serious somatic diseases (e.g., diabetes, hypertension, malignancies)?		
Yes	50	11.3
No	393	88.7
Are you or have you been suffering from any mental disorders (e.g., depression, neurosis, eating disorder)?		
Yes	135	30.3
No	308	69.5
In connection with the diagnosed mental disorder, are you currently receiving psychotherapy?		
Yes	41	9.3
No	402	90.7
Have you ever attempted to commit suicide?		
Yes	48	10.8
No	395	89.2
In connection with the diagnosed mental disorder, are you currently taking medication?		
Yes	50	11.3
No	393	88.7
Since the introduction of the pandemic-related constraints, have you observed any suicidal thoughts in you?		
Yes	47	10.6
No	396	89.4

**Table 4 ijerph-17-07015-t004:** The state of mental health and the level of perceived stress in the study group (*N* = 443).

Variables	M	SD	Min.	Max.
GHQ A somatic symptoms (e.g., headaches, exhaustion, weakness, subjective malaise)	9.12	4.78	0	21
GHQ B level of anxiety and insomnia	9.04	5.85	0	21
GHQ C social impairment	9.31	4.46	0	21
GHQ D depression	5.25	5.23	0	21
GHQ TOTAL	32.74	16.93	2	82
The Perceived Stress Scale (PSS-10)	18.99	6.63	2	30

GHQ—General Health Questionnaire-28; PSS-10—The Perceived Stress Scale; M—mean; SD—standard deviation; Min.—minimum value; Max.—maximum value.

**Table 5 ijerph-17-07015-t005:** Comparison of PSS 10 and MINI-COPE results among the respondents with high and low scores in the GHQ-28 questionnaire.

	GHQ > Sten Score 7 *N* = 234		GHQ < Sten Score 7 *N =* 210		*t*	*p*	*d*
	M	SD	M	SD			
PSS 10							
TOTAL	23.03	4.41	14.43	5.73	17.75	0.001	1.70
MINI−COPE							
Active coping	3.15	1.57	3.28	1.67	−0.79	0.430	0.08
Planning	3.74	1.48	3.86	1.64	−0.81	0.418	0.08
Positive reframing	3.21	1.71	3.98	1.72	−4.72	0.001	0.45
Acceptance	4.33	1.35	5.05	1.13	−6.05	0.001	0.58
Humour	2.24	1.25	2.48	1.36	−1.89	0.060	0.18
Religion	1.56	1.90	1.45	1.85	0.64	0.522	0.06
Use of emotional support	3.82	1.76	3.70	1.83	0.76	0.450	0.01
Use of instrumental support	3.53	1.68	3.04	1.79	2.99	0.002	0.29
Self distraction	3.74	1.36	3.88	1.48	−1.04	0.298	0.10
Denial	1.01	1.40	0.56	0.95	3.96	0.001	0.38
Venting	3.62	1.42	2.74	1.58	6.15	0.001	0.59
Substance use	1.01	1.54	0.51	1.11	3.90	0.001	0.36
Behavioral disengagement	1.98	1.60	0.82	1.19	8.54	0.001	0.76
Self blame	2.19	1.62	0.92	1.18	9.35	0.001	0.91

GHQ—General Health Questionnaire-28; PSS-10—The Perceived Stress Scale; M—mean; SD—standard deviation; t—Student’s *t*-test; p—materiality level.

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
