# Peer review of "Mental Health and the SARS-COV-2 Epidemic—Polish Research Study"

_ijerph, 2020, doi:10.3390/ijerph17197015_

Round 1
Reviewer 1 Report
1) Line 138 is not readable
2) Terms like "somatic symptoms" etc. in Tables, relating to questionaires 1 - 3 (Lines 103-114) must be more clearly described in the paper.
3) It is better wto write "chi-square" as a symbol not with multiple letters as word.
4) It is not clear if the outcomes are just from the currently performed research or there were the same research before COVID in 2019. And then those people were asked the same questions after COVID.
5) The papers looks as written in a great hurry, in not well organised form. It must be re-read, unified symbolic, terms, text emphasis, etc.
6) Data sample should be described separately.
7) Methodology should not be mixed with results.
8) Results should be clearly reported in separate section (again, do not mix matrhodology parts with data description parts with results parts). Keep the reseoning neat way.
Author Response
Reviewer 1.
Thank you for the detailed review of the article. We have marked all changes in the text.
Regarding the following comments:
Ad. 1. Line 138 is not readable.
Sentences: „The total result in the study group for the GHQ-28 questionnaire is M=32,74. This value corresponds to a sten score of 7 (high score) according to Polish standards, which means that the respondents are currently in a worse mental state than the population standard” (lines 137-138) was changed into:
„The total result in the study group for the GHQ-28 questionnaire is M=32,74 (7 sten, high score). It means that the respondents assess their mental health as unsatisfactory” (now lines 135-137)
Ad. 2. Terms like "somatic symptoms" etc. in Tables, relating to questionaires 1 - 3 (Lines 103-114) must be more clearly described in the paper.
The term „somatic symptoms” is more described (line 104 and 132 now).
Ad. 3. It is better to write "chi-square" as a symbol not with multiple letters as word.
We used χ2 symbol in throughout the text.
Ad 4. It is not clear if the outcomes are just from the currently performed research or there were the same research before COVID in 2019. And then those people were asked the same questions after COVID.
„The presented results refer to the first stage of the study carried out in April 2020. The second part of the study – scheduled for June and September 2020 – will assess the long-term effects of the pandemic that we are still facing” (Line 85-86).
In the same study we evaluated the frequency of alcohol consumption in the studied group:
Chodkiewicz, J.; Talarowska, M.; Miniszewska, J.; Nawrocka, N.; Biliński, P. Alcohol consumption reported during the COVID-19 pandemic: the initial stage. Int J Environ Res Public Health. 2020, 17(13), 4677.
Ad. 5. The papers looks as written in a great hurry, in not well organised form. It must be re-read, unified symbolic, terms, text emphasis, etc.
We carefully read the work once again and corrected the formal errors.
Ad. 6. Data sample should be described separately.
Ad. 7. Methodology should not be mixed with results.
Ad. 8. Results should be clearly reported in separate section (again, do not mix matrhodology parts with data description parts with results parts). Keep the reseoning neat way.
We applied to the Reviewer's comments (p. 3-5).
Reviewer 2 Report
It is clear that the COVID-19 crisis is related to numerous mental problems, so I think that the introduction should better expand and justify the importance of this study. For example, to go even deeper into the relationship between deteriorated mental health and stress in certain situations.
In Figure 1, the last line of the Psychological consequences box is not visible.
On the other hand, I think the second objective of the study is not appropriate for the study methodology. In a cross-sectional study, it is difficult to examine the possible risk factors associated with a certain variable. In addition, only comparison analysis was performed on the results, without carrying out logistic regressions that would allow determining the risk factors.
Starting from line 101 that describes the reliability of the instruments, it should be placed in the Material section. And tables 1, 2, and 3, as well as their respective descriptions, should be moved to the Results section.
In relation to the discussion and in line with what is explained in the methodology, I believe that the paragraph that begins on line 254 (In an attempt to determine the risk factors for the deterioration ...) should not be included since with the analyzes performed risk factors could not be determined.
Finally, I consider that the study has more limitations than those mentioned, for example, some of them:
- It is a cross-sectional study so it does not allow causal conclusions.
- The results cannot be extrapolated to another population, since the sample is Polish.
- The snowball sampling used is non-probabilistic, so it prevents a greater control of the sampling bias.
Author Response
Reviewer 2.
Thank you for the detailed review of the article. We have marked all changes in the text.
Regarding the following comments:
Ad 1. It is clear that the COVID-19 crisis is related to numerous mental problems, so I think that the introduction should better expand and justify the importance of this study. For example, to go even deeper into the relationship between deteriorated mental health and stress in certain situations.
Due to the fact that the article is a research study, we decided to put more emphasis on the obtained results, their analysis and discussion. At the present time, there are many publications relating to the theoretical relationship between mental health and the COVID-19 pandemic.
Ad. 2. In Figure 1, the last line of the Psychological consequences box is not visible.
The Figure 1 was corrected.
Ad 3. On the other hand, I think the second objective of the study is not appropriate for the study methodology. In a cross-sectional study, it is difficult to examine the possible risk factors associated with a certain variable. In addition, only comparison analysis was performed on the results, without carrying out logistic regressions that would allow determining the risk factors.
The aim of the study was changed (the abstract and the introduction - lines 80-81).
Ad 4. Starting from line 101 that describes the reliability of the instruments, it should be placed in the Material section. And tables 1, 2, and 3, as well as their respective descriptions, should be moved to the Results section.
We applied to the Reviewer's comments (p. 3-5).
Ad. 5. In relation to the discussion and in line with what is explained in the methodology, I believe that the paragraph that begins on line 254 (In an attempt to determine the risk factors for the deterioration ...) should not be included since with the analyzes performed risk factors could not be determined.
In our opinion, the indicated methods of statistical analysis have been adequately applied and the obtained results are statistically significant.
Ad 6. Finally, I consider that the study has more limitations than those mentioned, for example, some of them:
- It is a cross-sectional study so it does not allow causal conclusions.
- The results cannot be extrapolated to another population, since the sample is Polish.
- The snowball sampling used is non-probabilistic, so it prevents a greater control of the sampling bias.
Limitations section was extended by the comments of the reviewer.
Reviewer 3 Report
The objectives of the paper were 1) to investigate the effect of covid19 shutdown on the mental health of Polish adults and 2) identify possible risk factors for the associated emotional state. The results were from the first wave of survey that is part of a longitudinal study aiming to assess the long-term impact of the pandemic on the mental health and alcohol consumption among Polish adults. The manuscript is relevant and potentially useful, particularly the risk factors and coping mechanisms that the authors identified. I only have one major comment and a few minor comments.
Major comment:
There were a lot more women responded to the surveys compared to men. In addition, women and men tend to respond and cope to stressful events differently. I would be curious to see what the results might look like when analysis is stratified by gender.
Minor comments:
I spotted a few spelling errors throughout the manuscript so it would be good to edit the manuscript a few more times to check for those.
Abstract: p1 line 15: “Mini-Copre” instead of “Mini-Cope” ; p1 line 26: “Conslucions”;
The numbers in the conclusion seem unnecessary.
Introduction
Fig 1: the psychological consequences box appears to miss the word “violence”
P5 line 84: use “longitudinal study” instead of “elongated study”
Methods: In this section, I would appreciate a bit more details about the recruitment process. How did you advertise the online survey? What outlets did you use? What do you mean by the ‘snowball’ method in this context? Did the participants recruit other participants in the study? Was there any incentive for the participants?
Results
P6 line 136: typo M = 32,74
Discussion
P9, line 262: What are the strengths of your study?
P9, line 266: the use of the “summary and conclusions” heading might not be necessary as it might cause confusion with the actual conclusion section. The text reads fine to me without the heading.
The discussion is interesting.
Conclusions: I have not read a conclusion of a scientific study that uses bullet points (even though I like them). Most people condense the sentences into a single paragraph.
Author Response
Reviewer 3.
Thank you for the detailed review of the article. We have marked all changes in the text.
Regarding the following comments:
Ad. 1. There were a lot more women responded to the surveys compared to men. In addition, women and men tend to respond and cope to stressful events differently. I would be curious to see what the results might look like when analysis is stratified by gender.
We are aware of these differences. However, due to the breadth of the topic, we decided to prepare another publication about coping with men and women in the epidemic.
Ad 2. I spotted a few spelling errors throughout the manuscript so it would be good to edit the manuscript a few more times to check for those.
Abstract: p1 line 15: “Mini-Copre” instead of “Mini-Cope” ; p1 line 26: “Conslucions”;
The numbers in the conclusion seem unnecessary.
Introduction
Fig 1: the psychological consequences box appears to miss the word “violence”
P5 line 84: use “longitudinal study” instead of “elongated study”
Methods: In this section, I would appreciate a bit more details about the recruitment process. How did you advertise the online survey? What outlets did you use? What do you mean by the ‘snowball’ method in this context? Did the participants recruit other participants in the study? Was there any incentive for the participants?
Results
P6 line 136: typo M = 32,74
Discussion
P9, line 262: What are the strengths of your study?
P9, line 266: the use of the “summary and conclusions” heading might not be necessary as it might cause confusion with the actual conclusion section. The text reads fine to me without the heading.
The discussion is interesting.
Conclusions: I have not read a conclusion of a scientific study that uses bullet points (even though I like them). Most people condense the sentences into a single paragraph.
We have complied with all comments of the Reviewer.
Round 2
Reviewer 2 Report
Thanks for all the clarifications.
Good job after the changes.